# Learning to Learn Dense Gaussian Processes for Few-Shot Learning

**Ze Wang**[1][*]  **Zichen Miao**[1][*]  **Xiantong Zhen**[2,3]  **Qiang Qiu**[1]
Purdue University[1]    University of Amsterdam[2]    Inception Institute of Artificial Intelligence[3]
{zewang, miaoz, qqiu}@purdue.edu    x.zhen@uva.nl

## Abstract

Gaussian processes with deep neural networks demonstrate to be a strong learner for few-shot learning since they combine the strength of deep learning and kernels while being able to well capture uncertainty. However, it remains an open problem to leverage the shared knowledge provided by related tasks. In this paper, we propose to learn Gaussian processes with dense inducing variables by meta-learning for few-shot learning. In contrast to sparse Gaussian processes, we define a set of dense inducing variables to be of a much larger size than the support set in each task, which collects prior knowledge from experienced tasks. The dense inducing variables specify a shared Gaussian process prior over prediction functions of all tasks, which are learned in a variational inference framework and offer a strong inductive bias for learning new tasks. To achieve task-specific prediction functions, we propose to adapt the inducing variables to each task by efficient gradient descent. We conduct extensive experiments on common benchmark datasets for a variety of few-shot learning tasks. Our dense Gaussian processes present significant improvements over vanilla Gaussian processes and comparable or even better performance with state-of-the-art methods.

## 1   Introduction

Meta learning [31, 4, 18], also referred to as learning to learn, aims at learning to acquire shared knowledge from a set of related tasks so as to fast resolve novel tasks sampled from the same underlying task distribution. Meta learning largely stimulates the rise of few-shot learning [9], and the recent advantages are mainly driven by designing learning algorithms that learn from massive tasks and acquire prior knowledge, which combined with a small amount of labeled data, induces models that produce reliable predictions on novel tasks. In practice, the acquisition of prior knowledge can be realized in different forms. Model agnostic meta-learning (MAML) [9] learns to adapts to new tasks by few iterations of gradient descend, which inspires many follow-up methods [3, 12, 22, 46]. The adaptation of the entire network makes it hard to be scaled to large networks, and many recent efforts focus on adapting the last classification layer only [14, 5], while assuming a universal feature extractor that is shared across all tasks. Despite the remarkable progress, challenges remain due to the uncertainty in making predictions with very limited data, which requires models to have high robustness for few-shot learning.

Gaussian processes [25] serve as a powerful model for the inference of functions, which enjoy appealing properties including natural uncertainty quantification and robustness when the amount of data is limited. By combining the strong learning ability of deep neural networks, Gaussian processes with deep kernels further demonstrate improved performance on supervised learning [44]. To alleviate the computational cost, sparse Gaussian processes were widely studied by learning a sparse set of

---

[*]Equal contribution.

inducing variables to effectively approximate the full dataset. In sparse Gaussian processes, the inducing variables usually in the forms of pseudo data or a subset of training data are expected to capture statistics of the whole dataset.

Gaussian processes offer desirable robustness to data scarcity by capturing uncertainty, and enjoy the high expressiveness of deep kernels. This makes Gaussian processes a well-suited learning model for few-shot learning as demonstrated in recent works [33, 23, 38]. It is shown in deep kernel transfer (DKT) [23] that a Gaussian process with a meta-learned deep kernel can deliver strong performance through a simple kernel transfer. DKT relies on an important assumption that a universal deep kernel can be obtained through training on a limited amount of tasks. However, this assumption can be too restricted to generalize to novel tasks. In practice, when solving novel tasks, it is hardly guaranteed that the learned deep kernels can well explain the data due to the scarce labeled data, and failing to properly fit the Gaussian process prior can lead to inaccurate predictions. It is therefore crucial to leverage more knowledge to be shared among Gaussian processes for individual tasks, which remains an outstanding problem for Gaussian process few-shot learning.

In this paper, we introduce learning to learn Gaussian processes by a dense set of inducing variables for few-shot learning. In particular, we assume a Gaussian process prior over the prediction functions of few-shot learning tasks, which is specified by a set of inducing variables learned from data. These inducing variables are dense in the sense that they have a much larger cardinality compared to the support set of each individual task. Under the meta-learning setting, the dense inducing variables are learned to collect the shared knowledge from experienced tasks to improve the learning of new tasks effectively with limited data.

To achieve task-specific prediction functions, we propose to adapt the inducing variables to each task by the gradient descent update. Without adding extra model parameters, the adaptation permits online fitting to the Gaussian processes prior to each individual task, and therefore allows the model to better fit any novel tasks at test time. In addition, in contrast to sparse Gaussian processes, we learn the inducing variables in a low-dimensional deep feature space, which significantly reduces the compute cost of both learning and adapting the dense inducing variables.

The resultant *dense Gaussian processes* inherit the strong learning ability of deep kernels and enjoy the robustness to data scarcity and innate ability of uncertainty quantification, providing an effective few-shot learner. We validate the effectiveness of the proposed dense Gaussian processes on a variety of few-shot learning tasks. We observe that dense Gaussian processes achieve substantial performance improvements over vanilla deep Gaussian processes for few-shot learning and deliver state-of-the-art performance on common benchmarks for few-shot learning. Moreover, our dense Gaussian processes demonstrate strong generalization on cross-domain few-shot learning tasks.

In summary, we make three major contributions as follows:

- We introduce *dense Gaussian processes* by learning a set of dense inducing variables for few-shot learning. The inducing variables specify a Gaussian process prior over predictive functions across tasks, which collect shared knowledge from experienced tasks and provide strong inductive bias for learning new tasks efficiently and effectively.

- We propose gradient descent based adaptation to establish task-specific prediction functions based on the Gaussian process prior. The gradient descent-based adaptation is efficient without any auxiliary network components for inferring task representations.

- We conduct extensive experiments on common benchmark datasets for few-shot classification. The proposed dense Gaussian processes achieve consistent improvements over previous Gaussian processes with deep kernels and deliver comparable or even better results than state-of-the-art methods.

## 2 Methodology

In this section, we start with a brief review of few-shot learning; and the key ingredients that build the foundation of the proposed method. We then introduce in detail learning to learn Gaussian processes with dense inducing variables for few-shot learning.

## 2.1 Backgrounds

**Few-shot learning.** The common practice of training few-shot learners is to formulate it as a *episodic training*. In each episode, the training is realized by simulating a few-shot learning task, where the learner is provided with a set of few label data called the *support set* $\mathcal{S}$, and required to make predictions on the corresponding *query set* $\mathcal{Q}$ with unlabeled test samples. Taking few-shot image classification as an example, it is defined as a $N$-way $K$-shot learning problem, where $K$ is usually a small number, e.g., $K = 5$. Typically, in each episode indexed by $t$, one few-shot classification task is generated by first sampling $N$ categories from the training data, each of which contains $K$ samples to form the support set $\mathcal{S}_t = \{\mathbf{x}_1, \ldots, \mathbf{x}_{N \times K}\}$. An adaptation to the few-shot classification model is then performed on $\mathcal{S}_t$, by, e.g., computing prototypes [32], or updating the network parameters [9]. After adaptation, the query set is formed by evenly sampling $M$ samples from each category, i.e., $\mathcal{Q}_t = \{\mathbf{x}'_1, \ldots, \mathbf{x}'_{N \times M}\}$, to evaluate the updated model, and the error is propagated back to update the parameters.

**Gaussian processes with deep kernels.** Gaussian processes (GP) [25] place a prior distribution $p(\mathbf{f})$ over the target functions $\mathbf{f}$. Consider we have a set of $n$ datapoints, $\mathcal{D} = (\mathbf{x}_i, \mathbf{y}_i)_{i=1}^{N}$ with additive noise $\mathbf{y}_i = f(\mathbf{x}_i) + \epsilon_i$, where $\epsilon_i \sim \mathcal{N}(0, \sigma^2)$. The GP prior over $\mathbf{f}$ can be written as,

$$p(\mathbf{f}) = \mathcal{N}(\mathbf{0}, \mathcal{K}_{nn}), \tag{1}$$

where $\mathcal{K}_{nn} = \mathcal{K}(\mathbf{x}_n, \mathbf{x}_n)$ is the kernel matrix of $n$ training datapoints, and the likelihood is

$$p(\mathbf{y}|\mathbf{f}) = \prod_{i=1}^{n} p(\mathbf{y}_i|\mathbf{f}_i) = \prod_{i=1}^{n} \mathcal{N}(\mathbf{f}_i, \sigma^2). \tag{2}$$

Given a test data point $\mathbf{x}_*$, its predictive posterior is characterized by a Gaussian distribution, $\mathbf{f}_* \sim \mathcal{N}(\mathbf{f}_*|\mu_*, \sigma_*)$, where

$$\begin{aligned} \mu_* &= {\mathbf{k}_*}^T (\mathcal{K}_{nn} + \sigma^2 \mathbf{I})^{-1} \mathbf{y} \\ \sigma_* &= \mathbf{k}(\mathbf{x}^*, \mathbf{x}^*) - {\mathbf{k}_*}^T (\mathcal{K}_{nn} + \sigma^2 \mathbf{I})^{-1} \mathbf{k}_* \end{aligned} \tag{3}$$

Given a deep neural network $\Phi_\theta : \mathbf{X} \to \mathbf{Z}$, which maps from input space to deep feature space, we obtain the deep kernel function [44],

$$k(\mathbf{x}_i, \mathbf{x}_j | \theta) = k(\mathbf{\Phi}_\theta(\mathbf{x}_i), \mathbf{\Phi}_\theta(\mathbf{x}_j)) = k(\mathbf{z}_i, \mathbf{z}_j). \tag{4}$$

The dimensionality of deep feature space $\mathbf{Z}$ is usually smaller than the original input space $\mathbf{X}$, especially for image data, leading to more efficient kernel learning and inference. In this way, the GP prior in our method is parameterized by the network $\mathbf{\Phi}$ as $p(\mathbf{f}) = \mathcal{N}(\mathbf{f}|\mathbf{0}, \mathbf{K}_{nn})$, where $\mathbf{K}_{nn} = \mathbf{K}(\mathbf{z}_n, \mathbf{z}_n) = \mathbf{K}(\mathbf{x}_n, \mathbf{x}_n | \mathbf{\Phi}_\theta)$ is the covariance matrix calculated based on deep features, and the function values $\mathbf{f}$ are defined over deep features $\mathbf{f} = \mathbf{f}(\mathbf{Z}) = \mathbf{f}(\Phi(\mathbf{X}))$.

**Sparse Gaussian Processes** The posterior inference in a Gaussian Process induces a cubic complexity of $\mathcal{O}(n^3)$, which scales poorly with a huge amount of datapoints $n$. Sparse Gaussian processes [36, 34, 17] are widely studied to improve the scalability of standard GPs, and a popular direction introduces $m$ inducing variables $\{\tilde{\mathbf{Z}}_i\}_{i=1}^{m}$ to approximate the full GP marginal likelihood, where typical $m \ll n$ in favor of improved efficiency.

In sparse Gaussian processes, the optimization for the inducing variables $\{\tilde{\mathbf{Z}}_i\}_{i=1}^{m}$ can be achieved through variational inference [36]. And the GP prior [36] is augmented with the function values of inducing variables $\mathbf{f}_m \triangleq \mathbf{f}(\tilde{\mathbf{Z}})$ as $p(\mathbf{f}, \mathbf{f}_m) = p(\mathbf{f}|\mathbf{f}_m)p(\mathbf{f}_m)$, and the joint distribution can therefore be expressed as:

$$p(\mathbf{y}, \mathbf{f}, \mathbf{f}_m) = p(\mathbf{y}|\mathbf{f})p(\mathbf{f}|\mathbf{f}_m)p(\mathbf{f}_m), \tag{5}$$

where it is assumed that $\mathbf{y}$ is conditionally independent of $\mathbf{f}$ given $\mathbf{f}_m$. By minimizing the distance between the true posterior $p(\mathbf{f}, \mathbf{f}_m|\mathbf{y})$ and the variational approximate posterior $q(\mathbf{f}, \mathbf{f}_m) = p(\mathbf{f}|\mathbf{f}_m)q(\mathbf{f}_m)$, where $q(\mathbf{f}_m)$ is a free variational Gaussian distribution $\mathcal{N}(\mathbf{f}_m|\boldsymbol{\mu}, \boldsymbol{\Lambda})$, the maximization of log likelihood amounts to maximizing a variational lower bound:

$$\log p(\mathbf{y}) \geq \int_{\mathbf{f}, \mathbf{f}_m} q(\mathbf{f}, \mathbf{f}_m) \log \frac{p(\mathbf{y}, \mathbf{f}, \mathbf{f}_m)}{q(\mathbf{f}, \mathbf{f}_m)} d\mathbf{f} d\mathbf{f}_m \triangleq \mathcal{L}(\tilde{\mathbf{Z}}, \mathbf{\Phi}, q) \tag{6}$$

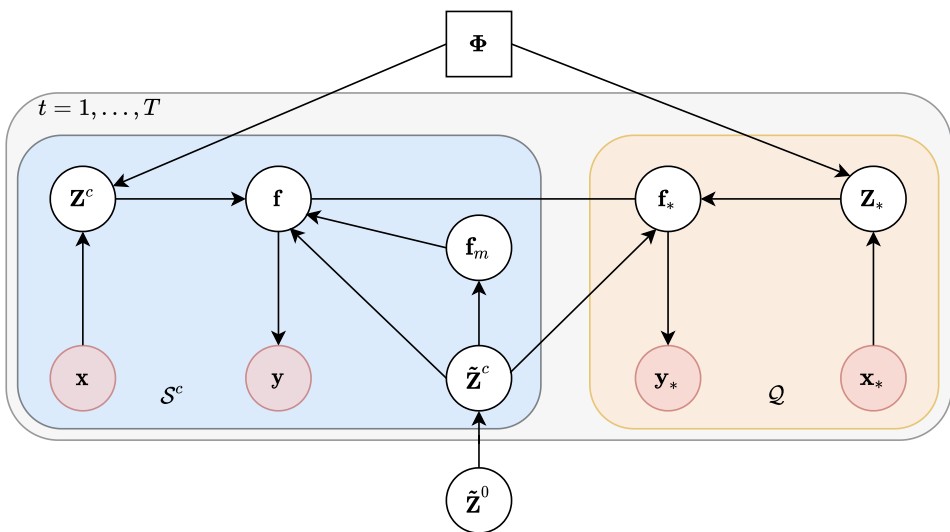

Figure 1: The graphical illustration of the proposed dense Gaussian processes for few-shot learning.

## 2.2 Learning to Learn Dense Gaussian Processes

In this section, we derive our dense Gaussian processes by inducing variables under the episodic training setting of meta-learning for few-shot learning.

**Adapting Gaussian Process Priors**   For a $C$-way $K$-shot classification task, we adapt $\mathbf{f}^c \sim$ $\text{GP}^c(0, \mathbf{K}^c)$, and the samples from the posterior predictive are combined based on a multi-class likelihood such as the softmax function. We would like to learn a Gaussian Process prior over the prediction function, which can be adapted to a few-shot learning task by its support set with a few gradient descent updates. The Gaussian process prior is established by a set of dense inducing variables, which is much larger than the support set in each task. The inducing variables are learned from data which serve to collect shared knowledge from experienced tasks to facilitate efficient learning of new tasks.

Formally, in an episode of the $C$-way $K$-shot task, the GP learner is provided with a support set $\mathcal{S}_t = \{\mathbf{x}_k^c, \mathbf{y}_k^c\}_{c=1,k=1}^{C,K}$. We adapt a task-specific GP prior in the form of specific kernels $\mathbf{K}^c(\Phi^c, \tilde{\mathbf{Z}}^c)$ for each class within the task, which can be achieved through an adaptation based on Eq. 6, with both parameters initialized by the meta representation $\Phi^0, \tilde{\mathbf{Z}}^0$,

$$\Phi^c(\Phi^0), \tilde{\mathbf{Z}}^c(\tilde{\mathbf{Z}}^0) = \arg\max_{\Phi, \tilde{\mathbf{Z}}} \mathcal{L}(\Phi, \tilde{\mathbf{Z}}, q). \tag{7}$$

As shown in [36], $\mathcal{L}(\Phi, \tilde{\mathbf{Z}}, q)$ can be first maximized by analytically solving the optimal $q^o(\mathbf{f}_m)$,

$$q^o(\mathbf{f}_m) = N(\mathbf{f}_m|\boldsymbol{\mu}^o, \boldsymbol{\Lambda}^o), \quad \boldsymbol{\mu}^o = \sigma^{-2}\mathbf{K}_{mm}\boldsymbol{\Sigma}\mathbf{K}_{mn}\mathbf{y}, \ \boldsymbol{\Lambda}^o = \mathbf{K}_{mm}\boldsymbol{\Sigma}\mathbf{K}_{mm}, \tag{8}$$

where now $n = |\mathcal{S}| = C \times K$, thus $\mathbf{K}_{mn}$ denotes the covariance matrix between $m$ inducing variables and $n$ support set data, and $\boldsymbol{\Sigma} = (\mathbf{K}_{mm} + \sigma^{-2}\mathbf{K}_{mn}\mathbf{K}_{mn}^T)^{-1}$. The Derivation is provided in the Appendix A. Thus, the lower bound can be written as,

$$\mathcal{L}(\Phi, \tilde{\mathbf{Z}}) = \log \mathcal{N}(\mathbf{y}|\mathbf{0}, \sigma^2 I + \hat{\mathbf{K}}_{nn}) - \frac{1}{2\sigma^2}\text{Tr}(\mathbf{K}_{nn} - \hat{\mathbf{K}}_{nn}), \tag{9}$$

where $\hat{\mathbf{K}}_{nn} = \mathbf{K}_{nm}\mathbf{K}_{mm}^{-1}\mathbf{K}_{nm}^T$. Note that (9) can be adopted for multi-class classification on the support set. To be specific, consider a binary classification task where $\mathbf{y} \in \{0, 1\}$, it can be solved by the proposed dense GP regression by assuming a Gaussian likelihood. With the *one-versus-rest* scheme, the regression model can be extended to solve the $C$-way classification task. Specifically, a regressor is learned for each class to predict whether a sample $\mathbf{x}$ belongs to this class (with $\mathbf{y} = 1$) or not (with $\mathbf{y} = 0$). Each regressor is formulated as a single dense GP with its own adapted inducing variables. By maximizing (9), we can obtain an adapted GP prior for each class that parameterized by

---

**Algorithm 1** Learning to learn dense inducing variables.

---

1: **Given**: A description of the tasks as $N$-way $K$-shot with $M$ query samples at each episode.
2: **Given**: Inner-loop and out-loop learning rates $\alpha$ and $\lambda$, number of steps $I$ for inner-loop optimization.
3: Initialize $\mathbf{\Phi}$ and $\tilde{\mathbf{Z}}$
4: **repeat**
5:     Sample task with support set $\mathcal{S} = \{\mathbf{x}_k^c\}_{c=1,k=1}^{C,K}$ and query set $\mathcal{Q} = \{\mathbf{x}_j'\}_{j=1}^M$ in each task.
6:     Extract feature vectors for both $\mathcal{S}$ and $\mathcal{Q}$ by applying $\mathbf{z}_k^c = \mathbf{\Phi}(\mathbf{x}_k^c)$ and $\mathbf{z}_j' = \mathbf{\Phi}(\mathbf{x}_j')$, respectively.
7:     **for** Class $n = 1 : N$ **do**
8:         **for** Inner iterations $i = 1 : I$ **do**
9:             Perform inner-loop adaptation and tuned the per-class inducing variables $\tilde{\mathbf{Z}}^{c,i}$ for a class-adapted GP prior by applying (14).
10:         **end for**
11:     **end for**
12:     With $\tilde{\mathbf{Z}}^{c,I}$ obtained, calculate the predictive posterior using (11), and perform the outer-loop adaptation by applying (15).
13: **until** Converge
14: **Return** $\mathbf{\Phi}$ and $\tilde{\mathbf{Z}}^0$.

---

$\tilde{\mathbf{Z}}^c, \mathbf{\Phi}^c$. Ideally, the well-trained inducing variables can sparsely approximate the full dataset, and reduce the prohibitive computation complexity to $\mathcal{O}(nm^2)$.

Given a query sample $\mathbf{x}_*$, the posterior predictive distribution for class $n$, $q(\mathbf{f}_*^c|\mathbf{x}_*^c) = \mathcal{N}(\mu_*^c, (\sigma^2)_*^c)$ can be calculated as

$$
\begin{aligned}
\mu_*^c &= \mathbf{k}_{*m}\mathbf{\Sigma}\mathbf{K}_{mn}^c\mathbf{y} \\
(\sigma^2)_*^c &= k^c(\mathbf{x}_*, \mathbf{x}_*) - \mathbf{K}_{*m}^c(\mathbf{K}_{mm}^c)^{-1}(\mathbf{k}_{*m}^c)^\top + \mathbf{k}_{*m}^c\mathbf{\Sigma}(\mathbf{k}_{*m}^c)^\top,
\end{aligned}
\tag{10}
$$

and the multi-class posterior predictive can be expressed as,

$$
p(\mathbf{y}_*^c = 1|\mathbf{x}_*) = \frac{\exp(\mathbf{f}_*^c \cdot \tau)}{\sum_{c'} \exp(\mathbf{f}_*^{c'} \cdot \tau)}, \quad \text{with } \mathbf{f}_*^c \sim q(\mathbf{f}_*^c|\mathbf{x}_*),
\tag{11}
$$

where $\tau$ is a learned temperature for MLE. Note the trainable temperature $\tau$ here allows automatic scaling to the unnormalized multi-class posterior predictive, and a similar implementation can also be found in [5]. Although sharing the same formulation, the $\tau$ here serves a completely different role as the temperature used in uncertainty calibration. The loss of empirical risk on the query set, acting as the outer-loop criteria, can be then be calculated as

$$
\mathcal{L}_{\text{Outer}}(\tilde{\mathbf{Z}}^0, \mathbf{\Phi}^0) = \sum_{(\mathbf{x}_*, \mathbf{y}_*) \in \mathcal{Q}} -\log p(\mathbf{y}_*|\mathbf{x}_*).
\tag{12}
$$

In this way, the GP prior adapted to the task $t$ by $\mathcal{S}_t$ is further supervised by the query set $\mathcal{Q}_t$. This in turn incorporates more knowledge from each task to consolidate task-shared meta knowledge into $\mathbf{\Phi}^0, \tilde{\mathbf{Z}}^0$, which enables new tasks to be learned efficiently.

**Dense Inducing Variables**    The aforementioned few-shot episodic training deep kernel allows an inner-loop adaptation to the model, and achieves better adaptation to the GP prior on novel tasks. However, performing adaptation to the entire deep kernel involves updating the deep neural network that parameterized the kernel, which introduces dramatically high computation cost and latency. Meanwhile, explicitly leveraging the meta knowledge learned from massive tasks, e.g., with memory that allows adaptive information retrieval [48], or data hallucination [45] that expands the support set of current tasks have demonstrated impressive results on few-shot learning. Inspired by the idea of explicit knowledge representation in few-shot learning, and data approximation through inducing variables in sparse Gaussian processes, we introduce *dense inducing variables*, which are learned as a rich set of inducing variables that carry the meta-knowledge of the entire task distribution in the latent deep feature space, and permit efficient model adaptation to novel tasks.

Specifically, instead of acting as a sparse approximation to the data of any particular task, the dense inducing variables here specify a shared Gaussian process prior over prediction functions of all tasks, are introduce to support the entire tasks distribution, from which a potentially infinite number of tasks can be sampled from.

To achieve effective modeling of the cross task GP prior, we allow the dense inducing variables to be have a larger cardinality than the support set in a single task, i.e., $m \gg |\mathcal{S}|$. Notably, unlike standard sparse Gaussian processes that learn the dense inducing variables as pseudo inputs in raw data space of high dimension, we propose to directly learn inducing variables in the same latent space of the feature from the deep neural network. By doing so, learning inducing variables effectively bypasses the heavy computation cost, and also enables efficient online adaptation to the inducing variables.

With dense inducing variables specifying a hyperprior over predictive functions across tasks, and included as hyperparameters in the deep kernel $\mathbf{\Phi}$, we now introduce the online update, which induces an adapted Gaussian process prior over the prediction function for a specific task, through a gradient descent based adaptation to the dense inducing variables.

Specifically, given the support set of the task with support set $\mathcal{S}_t$, we perform adaptation of the Gaussian process prior by using the few labeled data, and rewrite (7) as

$$\mathcal{L}_{\text{Inner}}(\tilde{\mathbf{Z}}) = \log \mathcal{N}(\mathbf{y}|\mathbf{0}, \sigma^2 I + \hat{\mathbf{K}}_{nn}) - \frac{1}{2\sigma^2}\text{Tr}(\mathbf{K}_{nn} - \hat{\mathbf{K}}_{nn}), \tag{13}$$

and adapt the inducing variables by

$$\tilde{\mathbf{Z}}^{c,i} = \tilde{\mathbf{Z}}^{c,i-1} - \alpha\nabla_{\tilde{\mathbf{Z}}^{c,i-1}}\mathcal{L}_{\text{Inner}}(\tilde{\mathbf{Z}}^{c,i-1}), \tag{14}$$

where $\alpha$ specifies an inner-loop learning rate, and each $\tilde{\mathbf{Z}}_n^0$ initialized with $\tilde{\mathbf{Z}}^0$. Thanks to the first-order approximation to the computationally heavy second-order derivative, the training with (14) can be remarkably accelerated.

Given the adapted inducing variables on the $i$-th task, the predictive posterior of function values on the query set can then be efficiently obtained by performing meta updates to both the feature extractor and the dense inducing variables calculated based on (12),

$$\begin{aligned}\mathbf{\Phi} &= \mathbf{\Phi} - \lambda\nabla_{\mathbf{\Phi}}\mathcal{L}_{\text{Outer}}(\mathbf{\Phi}, \tilde{\mathbf{Z}}^0) \\ \tilde{\mathbf{Z}}^0 &= \tilde{\mathbf{Z}}^0 - \lambda\nabla_{\tilde{\mathbf{Z}}^0}\mathcal{L}_{\text{Outer}}(\mathbf{\Phi}, \tilde{\mathbf{Z}}^0),\end{aligned} \tag{15}$$

with $\lambda$ specifying the learning rate for the outer loop.

Note that we assume a universal feature extractor $\mathbf{\Phi}$ and update it only in the outer loop, which is commonly used in meta-learning for efficiency and generalization to more powerful backbones. Meanwhile, the specification of the task-specific Gaussian process prior through adapting dense inducing variables endows our model with high flexibility and strong generalization to novel tasks without scarifying efficiency, as we will demonstrate in our experiments.

In contrast to the previous work that simply learns transferable Gaussian processes on few-shot learning [23], our method adopts episodic learning to adapt the Gaussian process prior to each task by updating the dense inducing variables, which parameterize the deep kernel jointly with the deep feature extractor $\mathbf{\Phi}$. The overall training procedure of the dense Gaussian process model is summarized in Algorithm 1.

## 3 Experiments

In this section, we support the proposed methods with strong experimental results. We present extensive experiments on both in-domain and cross-domain few-shot image classification. Visualizations of the uncertainty quantification with examples of few-shot regression are provided in Appendix Figure A. We finalize the discussions with the ablation study on the effectiveness of important components of the proposed method.

### 3.1 Datasets and Settings

We perform experiments on the widely used few-shot learning benchmarks including *mini*ImageNet, *tiered*ImageNet, CIFAR-FS , and Caltech-UCSD [42] (CUB). In *mini*ImageNet [41], there are 100

Table 1: Performance comparisons on *mini*ImageNet and the Conv-4 network with 95% confidence intervals.

| Methods | *mini*ImageNet | | *tiered*ImageNet | | CIFAR-FS | |
|---|---|---|---|---|---|---|
| | 1-shot | 5-shot | 1-shot | 5-shot | 1-shot | 5-shot |
| MatchingNets [41] | $43.56 \pm 0.84$ | $55.31 \pm 0.73$ | - | - | - | - |
| MAML [10] | $48.70 \pm 1.84$ | $63.11 \pm 0.92$ | $51.67 \pm 1.81$ | $70.30 \pm 1.75$ | $58.90 \pm 1.91$ | $71.52 \pm 1.10$ |
| Reptile [22] | $49.97 \pm 0.32$ | $65.99 \pm 0.58$ | - | - | - | - |
| R2-D2 [6] | $48.70 \pm 0.60$ | $65.50 \pm 0.60$ | - | - | $60.20 \pm 1.80$ | $70.91 \pm 0.91$ |
| VERSA [14] | $53.31 \pm 1.80$ | $67.30 \pm 0.91$ | - | - | $62.51 \pm 1.70$ | $75.11 \pm 0.91$ |
| RelationNets [35] | $50.44 \pm 0.82$ | $65.32 \pm 0.70$ | $54.48 \pm 0.93$ | $65.32 \pm 0.70$ | $55.00 \pm 1.01$ | $69.30 \pm 0.80$ |
| ProtoNets [32] | $47.40 \pm 0.60$ | $65.41 \pm 0.52$ | $53.31 \pm 0.89$ | $72.69 \pm 0.74$ | $55.50 \pm 0.70$ | $72.01 \pm 0.60$ |
| VSM [48] | $54.73 \pm 1.60$ | $68.01 \pm 0.90$ | $56.88 \pm 1.71$ | $74.65 \pm 0.81$ | $63.42 \pm 1.90$ | $77.93 \pm 0.80$ |
| DKT [23] | $49.73 \pm 0.07$ | $64.00 \pm 0.09$ | - | - | - | - |
| OVE PG GP + Cosine (ML) [33] | $50.02 \pm 0.35$ | $64.58 \pm 0.31$ | - | - | - | - |
| OVE PG GP + Cosine (PL) [33] | $48.00 \pm 0.24$ | $67.14 \pm 0.23$ | - | - | - | - |
| **Our** | $\mathbf{56.32} \pm 0.28$ | $\mathbf{72.64} \pm 0.26$ | $\mathbf{58.43} \pm 0.38$ | $\mathbf{76.17} \pm 0.34$ | $\mathbf{64.17} \pm 0.31$ | $\mathbf{78.42} \pm 0.26$ |

Table 2: Performance comparisons on the CUB dataset with 95% confidence intervals.

| Methods | CUB (Conv-4) | | CUB (ResNet-10) | |
|---|---|---|---|---|
| | 1-shot | 5-shot | 1-shot | 5-shot |
| Feature Transfer | $46.19 \pm 0.64$ | $68.40 \pm 0.79$ | $63.64 \pm 0.91$ | $81.27 \pm 0.57$ |
| ABML [26] | $49.57 \pm 0.42$ | $68.94 \pm 0.16$ | - | - |
| Baseline ++ [7] | $61.75 \pm 0.95$ | $78.51 \pm 0.59$ | $69.55 \pm 0.89$ | $85.17 \pm 0.50$ |
| MatchingNet [41] | $60.19 \pm 1.02$ | $75.11 \pm 0.35$ | $71.29 \pm 0.87$ | $83.47 \pm 0.58$ |
| ProtoNet [32] | $52.52 \pm 1.90$ | $75.93 \pm 0.46$ | $73.22 \pm 0.92$ | $85.01 \pm 0.52$ |
| RelationNet [35] | $62.52 \pm 0.34$ | $78.22 \pm 0.07$ | $70.47 \pm 0.99$ | $83.70 \pm 0.55$ |
| MAML [10] | $56.11 \pm 0.69$ | $74.84 \pm 0.62$ | $70.32 \pm 0.99$ | $80.93 \pm 0.71$ |
| Bayesian MAML [47] | $55.93 \pm 0.71$ | $72.87 \pm 0.26$ | - | - |
| DKT [23] | $62.96 \pm 0.62$ | $77.76 \pm 0.62$ | $72.27 \pm 0.30$ | $85.64 \pm 0.29$ |
| OVE PG GP + Cosine (ML) [33] | $63.98 \pm 0.43$ | $77.44 \pm 0.18$ | - | - |
| OVE PG GP + Cosine (PL) [33] | $60.11 \pm 0.26$ | $79.07 \pm 0.05$ | - | - |
| **Ours** | $\mathbf{69.18} \pm 0.41$ | $\mathbf{81.48} \pm 0.58$ | $\mathbf{78.83} \pm 0.67$ | $\mathbf{89.97} \pm 0.63$ |

image classes from a subset of ImageNet [8], with 600 images for each class. We follow the standard practice [9] to split the training, validation, and testing sets with 64, 16, and 20 classes, respectively. *tiered*ImageNet [28] is a large subset of ImageNet that contains 608 classes with 1,300 samples in each class. Specifically, in *tiered*ImageNet, there are 351 classes from 20 categories for training, 97 classes from 6 categories for validation, and 160 classes from 8 different categories for testing. Samples for both *mini*ImageNet and *tiered*ImageNet are random cropped and resized to $84 \times 84$ for training, and standard center cropping is performed to the testing images. The 200 classes in the CUB dataset is divided into 100, 50, and 50 classes, for training, validation, and testing, respectively. CIFAR-FS adopts all the 100 classes in the CIFAR-100 dataset with training, validation, and testing splits of 64, 16, and 20 classes, respectively. Each class contains 600 image samples. The image resolution for CIFAR-FS is $32 \times 32$.

## 3.2 Few-Shot Image Classification

The results of few-shot classification on *mini*ImageNet, *tiered*ImageNet, CIFAR-FS , and Caltech-UCSD (CUB) are reported in Tables 1 and 2. For comprehensive comparisons, we adopt two backbone networks with different scales. On all three datasets, we achieve comparable or often better performance than state-of-the-art methods. Our method yields the accuracy of $72.64$ on the *mini*ImageNet dataset under the 5-way 5-shot setting, which surpasses the previous best method by up to $5\%$. In particular, our dense Gaussian processes achieve much better performance than the recent Gaussian processes based methods [23, 33]. This demonstrates the effectiveness of our method by learning dense inducing variables. For more experimental detail please refer to Appendix Section B.

Table 3: Performance comparisons for cross-domain few-shot classification with 95% confidence intervals.

| Methods | Omniglot → EMNIST | | miniImageNet → CUB | |
| --- | --- | --- | --- | --- |
| | 1-shot | 5-shot | 1-shot | 5-shot |
| Feature Transfer | $64.22 \pm 1.24$ | $86.10 \pm 0.84$ | $32.77 \pm 0.35$ | $50.34 \pm 0.27$ |
| ABML [26] | $76.37 \pm 0.29$ | $87.96 \pm 0.28$ | $29.35 \pm 0.26$ | $45.74 \pm 0.33$ |
| Baseline ++ [7] | $56.84 \pm 0.91$ | $80.01 \pm 0.92$ | $39.19 \pm 0.12$ | $57.31 \pm 0.11$ |
| MatchingNet [40] | $75.01 \pm 2.09$ | $87.41 \pm 1.79$ | $36.98 \pm 0.06$ | $50.72 \pm 0.36$ |
| ProtoNet [32] | $72.04 \pm 0.82$ | $87.22 \pm 1.01$ | $33.27 \pm 1.09$ | $52.16 \pm 0.17$ |
| RelationNet [35] | $75.62 \pm 1.00$ | $87.84 \pm 0.27$ | $37.13 \pm 0.20$ | $51.76 \pm 1.48$ |
| MAML [10] | $72.68 \pm 1.85$ | $83.54 \pm 1.7$ | $34.01 \pm 1.25$ | $48.83 \pm 0.62$ |
| Bayesian MAML [47] | $63.94 \pm 0.47$ | $65.26 \pm 0.30$ | $33.52 \pm 0.36$ | $51.35 \pm 0.16$ |
| DKT [23] | $75.40 \pm 1.10$ | $90.30 \pm 0.49$ | $40.14 \pm 0.18$ | $56.40 \pm 1.34$ |
| OVE PG GP (ML) [33] | $68.43 \pm 0.67$ | $86.22 \pm 0.20$ | $39.66 \pm 0.18$ | $55.71 \pm 0.31$ |
| OVE PG GP (PL) [33] | $77.00 \pm 0.50$ | $87.52 \pm 0.19$ | $37.49 \pm 0.11$ | $57.23 \pm 0.31$ |
| **Ours** | $\mathbf{78.32} \pm 0.49$ | $\mathbf{90.78} \pm 0.24$ | $\mathbf{43.45} \pm 0.38$ | $\mathbf{60.48} \pm 0.53$ |

### 3.3 Cross-Domain Few-Shot Image Classification

The efficient adaptation to the GP prior through updating the dense inducing variables enables the proposed method with improved generalization to novel tasks with unseen categories. To fully demonstrate the remarkable generalization, we perform cross-domain few-shot image classification. Following common practice, we adopt two cross-domain settings, Omniglot → EMNIST and miniImageNet→ CUB. The results are reported in Table 3. The proposed dense Gaussian processes demonstrate high generalizability across domains for few-shot classification and achieve much better performance than the recent Gaussian process based methods [23, 33] in the setting of generalization from Omniglot to CUB.

### 3.4 Ablation Study

We perform extensive ablation studies to provide insights into the effectiveness of the proposed dense Gaussian processes for few-shot learning. These experiments are performed on the CUB dataset.

**Effectiveness of dense inducing variables** The proposed dense inducing variables enable shared knowledge to be collected from experienced tasks and applied to new unseen tasks. We show how the number of inducing variables affects the performance of our model. As shown in Figure 2, we report performance on both 5-way 1-shot and 5-way 5-shot experiments. We test a sequence of values of $m$ starting from 8, which is just slightly larger than the number of support set samples in a 5-way 1-shot experiment. And the value of $m$ is progressively increased to 1024. It is clearly demonstrated that, on both experiments with different shots, the performance grows monotonically w.r.t. the number of inducing points that are utilized. And the performance tends to saturate when $m > 256$. To maintain a balance between efficiency and performance, we choose $m = 256$ throughout all the experiments. The results in Figure 2 also validate the effectiveness of specifying the GP prior through 'dense' inducing variables, with much more elements compared to the support set in each task.

**Effectiveness of inner-loop adaptation** In the proposed method, the task-specific GP prior adaptation is achieved by few-step gradient descent to the dense inducing variables. For the selection of the best configuration of the inner-loop optimization, we present comparisons performed with the number of steps $I$ in the inner-loop gradient descent, and the inner-loop learning rate $\alpha$. As shown in Figure 14, we plot the classification accuracy for 5-way 5-shot setting, and show the performance obtained by conducting 1 to 20 steps of inner-loop update with three different values of $\alpha$. It is clearly shown that the performance reaches the peak within 5 steps of inner-loop update, and it does not improve with even more steps of adaption, especially when combined with larger learning rates. This indicates the efficiency of the proposed adaption of the Gaussian process prior to each task. Moreover, a large $\alpha$ can lead to performance drop very fast when performing more than 1 step of inner-loop update. To keep the balance between accuracy and efficiency, we choose $\alpha = 0.01$ and $I = 5$ throughout all the experiments.

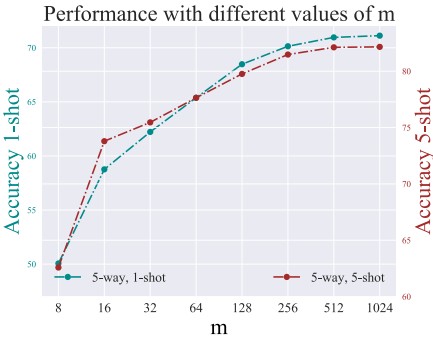
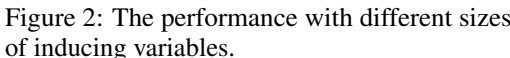
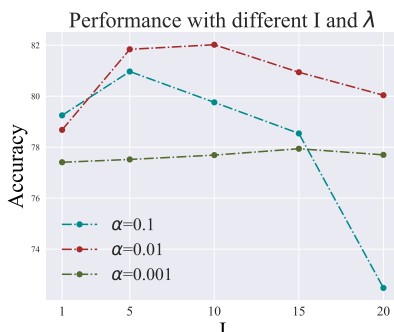

Figure 2: The performance with different sizes of inducing variables.

Figure 3: The performance with different numbers of adaptation steps and hyper-parameters.

## 4 Related Work

Metric learning [32] has been explored for few-shot learning, which assumes that different related tasks would share the same metric to measure similarity. Vinyals et al. [41] proposed the matching network, which learns to map a small labeled support set and an unlabelled example to its label, obviating the need for fine-tuning to adapt to new class types. This work was originally developed for one-shot learning, and extended to a few-shot setting by Snell et al. [32]. They proposed the prototypical network to learn a metric space in which classification can be performed by computing distances to prototype representations of each class and the prototype of each class is the cluster of samples in that class. To enhance the expressivity of the prototypes, Allen et al. [1] proposed infinite mixture prototypes to adaptively represent both simple and complex data distributions for few-shot learning. Satorras et al. [30] solved few-shot learning with the prism of inference on a partially observed graphical model.

Meta-learning based on optimization has been extensively studied [27, 10, 2]. The core idea of optimization-based methods is to learn an optimization procedure that is shared across tasks, which can be applied to new tasks for quick adaptation, once learned in the meta-train stage. Ravi et al. [27] proposed an LSTM based meta-learner to learn the exact optimization algorithm used to train a neural network classifier in the few-shot regime; in their methods, apart from the learned optimization algorithm, a good initialization of model parameters is also obtained after training. Finn et al. [11] proposed a model agnostic meta-learning (MAML) algorithm, which was built upon the assumption that related tasks sharing initial parameters of neural networks could be adapted to specific tasks with a few steps of gradient descent updates. It was believed that the initial weights combined with a few more steps of gradient descent can approximate any learning algorithm, and thus gradient-based meta-learning had a number of practical benefits. Finn et al. [13] extended MAML in a probabilistic framework. Rajeswaran et al. [24] developed the implicit MAML [10] algorithm, which depended only on the solution to the inner level optimization. Zintgraf et al. [50] updated context parameters with one or several gradient steps on a task-specific loss that serves as an additional input to the model and were adapted on individual tasks.

It has also been investigated to explicitly design a meta-learner to learn a base-learner. Bertinetto et al. [6] explored the feasibility of incorporating fast solvers with closed-form solutions as the base learning component of a meta-learning system. Gordon et al. [15] developed meta-learning approximate probabilistic inference for prediction. The support set was used to produce the parameter distribution of the classifier, which was applied to the query set for prediction. Mishra et al. [20] proposed a generic meta-learner architecture that used a novel combination of temporal convolutions and soft attention. Zhen et al. [49] introduced learning to learn kernels with variational random features for few-shot learning, where a specific kernel is inferred for each individual tasks conditioning on global context information collected by LSTMs. Recently, memory has generated increasing attention in the machine learning community, which was used to augment deep neural networks [43, 16]. It has also been introduced to the meta-learning framework for few-shot learning [29, 21, 48].

Gaussian processes have recently been introduced for Few-shot Learning [39, 23, 33, 33]. Tossou et al. [39] learn a parameterized kernel operator that can be combined with a differentiable kernel algorithm during inference. Deep Kernel Transfer (DKT) [23] explore Gaussian processes for few-shot classification, which learns covariance functions parameterized by deep neural networks. More recently, Titsias et al. [38] applies Gaussian processes to meta-learning by maximizing the mutual information between the query set and a latent representation of the support set. Snell et al. [33] develop a Gaussian process classifier by combining Pólya-Gamma augmentation and the one-vs-each softmax approximation [37] in order to efficiently marginalize over functions rather than model parameters. In our work, we explore a dense set of inducing variables in Gaussian processes for few-shot classification. The resultant dense Gaussian processes enable more knowledge to be shared among related tasks and transferred to new unseen tasks for efficient and effective learning. Notably, the idea of inducing points in GP also stimulates progress in many other directions of research. Set transformer [19] introduced a transformer architecture with an inducing-point-inspired design for efficient computation. This serves as a piece of supportive evidence for using inducing points in meta-learning, as we show in this paper that using inducing points to collect and accumulate shared knowledge from previously seen, related tasks offers a strong inductive bias for efficiently and effectively solving new tasks with few labeled data only.

## 5 Conclusion

In this paper, we introduce learning to learn dense inducing variables for few-shot learning. By specifying a Gaussian process prior prediction functions, we introduce dense inducing variables, which are learned from data to collect shared knowledge from previous work to facilitate efficient and effective learning of new tasks. With a universal deep feature extractor learned from the training tasks, we achieve task-specific Gaussian process prior by adapting the dense inducing variables through an inner-loop few-step gradient optimization. The resultant dense Gaussian processes are endowed with strong generalization and robustness. The effectiveness of the proposed method is validated on a variety of benchmark datasets, and state-of-the-art performance across tasks is observed. Notably, the substantial performance improvements on the cross-domain few-shot learning further demonstrate the strong generalization of the proposed method.

## 6 Acknowledgements

This work was supported by the DARPA TAMI program under No. HR00112190038.

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
