# Appendix

## A Derivative of ELBO

We first derive the (6). Starting from Kullback-Leibler divergence (KL divergence $D_{\text{KL}}$) between the variational approximate posterior $q(\mathbf{f}, \mathbf{f}_m)$ and the true posterior $p(\mathbf{f}, \mathbf{f}_m|\mathbf{y})$, $D_{\text{KL}}[q(\mathbf{f}, \mathbf{f}_m)||p(\mathbf{f}, \mathbf{f}_m|\mathbf{y})]$, we can derive that,

$$
\begin{aligned}
D_{\text{KL}}[q(\mathbf{f}, \mathbf{f}_m)||p(\mathbf{f}, \mathbf{f}_m|\mathbf{y})] &= \mathbb{E}_q[\log \frac{q(\mathbf{f}, \mathbf{f}_m)}{p(\mathbf{f}, \mathbf{f}_m|\mathbf{y})}] \\
&= \mathbb{E}_q[\log \frac{q(\mathbf{f}, \mathbf{f}_m)p(\mathbf{y})}{p(\mathbf{f}, \mathbf{f}_m, \mathbf{y})}] \\
&= \mathbb{E}_q[\log q(\mathbf{f}, \mathbf{f}_m) + \log p(\mathbf{y}) - \log p(\mathbf{f}, \mathbf{f}_m, \mathbf{y})] \geq 0.
\end{aligned}
\tag{1}
$$

Thus,

$$
\begin{aligned}
\log p(\mathbf{y}) &\geq \mathbb{E}_q[\log p(\mathbf{f}, \mathbf{f}_m, \mathbf{y}) - \log q(\mathbf{f}, \mathbf{f}_m)] \\
&= \mathbb{E}_q[\log \frac{p(\mathbf{f}, \mathbf{f}_m, \mathbf{y})}{q(\mathbf{f}, \mathbf{f}_m)}] \triangleq \mathcal{L}(\tilde{\mathbf{Z}}, \mathbf{\Phi}, q),
\end{aligned}
\tag{2}
$$

which is consistent with (6). $\mathcal{L}(\tilde{\mathbf{Z}}, \mathbf{\Phi}, q)$ can be further reduced as,

$$
\begin{aligned}
\mathcal{L}(\tilde{\mathbf{Z}}, \mathbf{\Phi}, q) &= \int_{\mathbf{f}, \mathbf{f}_m} p(\mathbf{f}|\mathbf{f}_m)q(\mathbf{f}_m) \log \frac{p(\mathbf{y}|\mathbf{f})p(\mathbf{f}|\mathbf{f}_m)p(\mathbf{f}_m)}{p(\mathbf{f}|\mathbf{f}_m)q(\mathbf{f}_m)} d\mathbf{f} d\mathbf{f}_m \\
&= \int_{\mathbf{f}_m} q(\mathbf{f}_m)\{ \int_{\mathbf{f}} p(\mathbf{f}|\mathbf{f}_m) \log p(\mathbf{y}|\mathbf{f}) d\mathbf{f} + \log \frac{p(\mathbf{f}_m)}{q(\mathbf{f}_m)} \} d\mathbf{f}_m \\
&= \int_{\mathbf{f}_m} q(\mathbf{f}_m)\{ \log \frac{L(\mathbf{f}_m, \mathbf{y})p(\mathbf{f}_m)}{q(\mathbf{f}_m)} \} d\mathbf{f}_m,
\end{aligned}
\tag{3}
$$

where $\log L(\mathbf{f}_m, \mathbf{y})$ is the lower bound of the likelihood $\log p(\mathbf{y}|\mathbf{f}_m)$, and $\log L(\mathbf{f}_m, \mathbf{y}) = \log \int_{\mathbf{f}} p(\mathbf{y}|\mathbf{f})p(\mathbf{f}|\mathbf{f}_m)d\mathbf{f} = \log \mathcal{N}(\mathbf{y}|\mathbf{K}_{nm}\mathbf{K}_{mm}^{-1}\mathbf{f}_m, \sigma^2 I) - \frac{1}{2\sigma^2} Tr(\hat{\mathbf{K}})$, with $\tilde{\mathbf{K}} = \mathbf{K}_{nn} - \hat{\mathbf{K}}_{nn} = \mathbf{K}_{nn} - \mathbf{K}_{nm}\mathbf{K}_{mm}^{-1}\mathbf{K}_{mn}$. By Jensen's inequality,

$$
\begin{aligned}
\mathcal{L}(\tilde{\mathbf{Z}}, \mathbf{\Phi}, q) &\leq \log \int_{\mathbf{f}_m} L(\mathbf{f}_m, \mathbf{y})p(\mathbf{f}_m)d\mathbf{f}_m \\
&= \log \int_{\mathbf{f}_m} \mathcal{N}(\mathbf{y}|\mathbf{K}_{nm}\mathbf{K}_{mm}^{-1}\mathbf{f}_m, \sigma^2 I)d\mathbf{f}_m - \frac{1}{2\sigma^2} Tr(\tilde{\mathbf{K}}) \\
&= \log \mathcal{N}(\mathbf{y}|\mathbf{0}, \sigma^2 I + \mathbf{K}_{nm}\mathbf{K}_{mm}^{-1}\mathbf{K}_{mn}) - \frac{1}{2\sigma^2} Tr(\mathbf{K}_{nn} - \hat{\mathbf{K}}_{nn}) \triangleq \mathcal{L}(\tilde{\mathbf{Z}}, \mathbf{\Phi}),
\end{aligned}
\tag{4}
$$

which matches (9).

## B Implementation Details

In this section, we provide implementation details regarding the proposed method.

All experiments are conducted on a server with 8 Nvidia RTX 3090 graphic cards, and each has 24GB memory. Every experiment we report can be trained and tested on a single card. The machine is also equipped with 512GB memory and two AMD EPYC 7502 CPUs. We use PyTorch for the implementations of all experiments.

Every experiment is trained with the ADAM optimizer, with a learning rate of $0.001$. We train all networks for a total of 80,000 iterations, with the learning rate decays at the 60,000-th and the 70,000-th iteration. In practice, to improve the stability of training, we clip the predicted $(\sigma^2)_*^c$ to values between 0.01 to 100.0.

**Network architectures.**

Table A: The architectures of CNNs.

| Conv-4 architecture | |
|---|---|
| **Output size** | **Layers** |
| 84×84×3 | Input images |
| 42×42×64 | *Conv* (3×3), *BatchNorm*, *ReLU*, *MaxPool* (2×2, stride=2) |
| 21×21×64 | *Conv* (3×3), *BatchNorm*, *ReLU*, *MaxPool* (2×2, stride=2) |
| 10×10×64 | *Conv* (3×3), *BatchNorm*, *ReLU*, *MaxPool* (2×2, stride=2) |
| 5×5×64 | *Conv* (3×3), *BatchNorm*, *ReLU*, *MaxPool* (2×2, stride=2) |
| 1600 | flatten |

| ResNet-12 | |
|---|---|
| **Output size** | **Layers** |
| 84×84×3 | Input images |
| 42×42×64 | [*Conv* (3×3), *BatchNorm*, *ReLU*] × 3, *MaxPool* (2×2, stride=2) |
| 21×21×128 | [*Conv* (3×3), *BatchNorm*, *ReLU*] × 3, *MaxPool* (2×2, stride=2) |
| 10×10×256 | [*Conv* (3×3), *BatchNorm*, *ReLU*] × 3, *MaxPool* (2×2, stride=2) |
| 5×5×512 | [*Conv* (3×3), *BatchNorm*, *ReLU*] × 3, *MaxPool* (2×2, stride=2) |
| 1×1×512 | *Global Average Pool* |
| 512 | flatten |

| ResNet-10 | |
|---|---|
| **Output size** | **Layers** |
| 224×224×3 | Input images |
| 56×56×3 | *Conv*(7×7, stride=2), *BatchNorm*, *ReLU*, *MaxPool*(3×3, stride=2) |
| 56×56×64 | [*Conv* (3×3), *BatchNorm*, *ReLU*] × 2, |
| 28×28×128 | *Conv* (3×3, stride=2), *BatchNorm*, *ReLU*,
*Conv* (3×3), *BatchNorm*, *Conv* (1×1, stride=2), *ReLU* |
| 14×14×256 | *Conv* (3×3, stride=2), *BatchNorm*, *ReLU*,
*Conv* (3×3), *BatchNorm*, *Conv* (1×1, stride=2), *ReLU* |
| 7×7×512 | *Conv* (3×3, stride=2), *BatchNorm*, *ReLU*,
*Conv* (3×3), *BatchNorm*, *Conv* (1×1, stride=2), *ReLU* |
| 1×1×512 | *Global Average Pool* |
| 512 | flatten |

Table B: Performance with different kernels. Numbers are obtained on 5-way 5-shot experiments on CUB with Conv-4 backbone.

| Kernel | Linear | Cosine | RBF |
|---|---|---|---|
| Accuracy | 80.38 | 81.48 | 77.62 |

**Kernel.** The pair-wise integration $\mathbf{k}(\cdot)$ is decided by the kernel function. Here we present the results obtained with different kernels in Table B. We adopt cosine kernels in all the experiments in Section 4.

## C  Few-shot Regression

We present in Figure A the results of few-shot regression with uncertainty visualized. The setting of the few-shot regression experiments follows exactly the one introduced in [9] with sine waves.

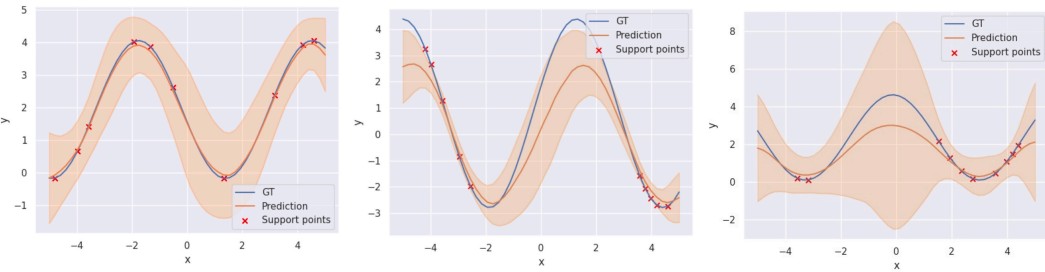

Figure A: Visualizations of the uncertainty quantification in few-shot regression experiments.