# OpenReview forum: "Learning to Learn Dense Gaussian Processes for Few-Shot Learning"
_NeurIPS.cc/2021/Conference — NeurIPS 2021 Poster_

### Official Review · Reviewer_jkmZ · 2021-06-29

**Rating:** 6
**Confidence:** 4

**Summary:**

This paper proposes an extension of the Gaussian process (GP) approach presented in [1] for the setup of few-shot learning. With each new task, the data is first processed by a neural network (NN) to obtain low-dimensional feature representations, and then a GP classifier acts on these features. Since the number of data points per task is limited, the authors suggest augmenting the model with inducing inputs that are shared among all tasks. Updating the parameters follows the standard meta-learning procedures (e.g., MAML). Using the lower bound presented in [1], an internal loop updates the task-specific inducing inputs. And by maximizing the predictive likelihood on the query set, an outer loop updates both the global inducing inputs and the network parameters. Experiments on standard FSL and cross-domain few-shot benchmarks show an improved accuracy over relevant baseline methods.

**Limitations And Societal Impact:**

The authors did not address the potential limitations and negative societal impact of their work.  I think that authors should address this in the rebuttal.
To the best of my judgment, there is no potential negative societal impact for this paper.

**Main Review:**

I think that using Gaussian processes with NNs for few-shot challenges is a promising direction. However, there are some issues I would like the authors to address before I can pass it.

This paper presents the following merits:
- The paper shows strong results in a variety of experiments and against many baselines.
- Using inducing variables to generate a richer dataset is a good idea.
- The paper is written clearly and is easy to follow besides several points that will be described next.

Some points that I found problematic:
- There are two crucial points that were not entirely clear to me, and since the code wasn't provided with the submission I could not get an answer myself. I will be happy to get clarification for both issues.
(1) This paper builds on [1] which was designed for regression tasks with Gaussian likelihoods. To use this method, it seems that the authors referred to the discrete class labels as if they are Gaussian distributed. If indeed I understood correctly, it does not seem like a natural thing to do. Can you please clarify that point and further explain the intuition.
(2) Unlike training in which the inducing inputs are jointly learned across tasks, when testing the model the classes are new and there should be no information sharing between tasks. Is that correct? If so, can you please clarify if and how does your method gain from sharing the inducing inputs of these new classes?  If the inducing location is only adjusted based on the inner loop iterations of the current task, does that suffice? I think that some analysis and more explanations regarding this part are required.
- Many implementation details are missing. For example, did you perform a grid search over hyperparameters (besides the ones shown in the ablation study)? what temperature did you use in the softmax? how many epochs of training did you use? etc. In general, the text should provide much more details about the datasets, the experimental setups, and the baseline methods.
- I think this paper omits important GPC references [2, 3, 4]. [2] handled a few-shot scenario as well, and all methods used inducing inputs in the embedding space of a NN.
- From my experience temperature scaling often impact significantly on the classifier performance (for example, see [5] for a recent study on this topic). If indeed you adjusted the temperature for your method, did you try to tune it for relevant baseline methods as well? Specifically, I think that the performance of the main competitor OVE-GP (PL) should also be examined under the same conditions.

Minor comments:
- There are some typing mistakes: gradient decend (L25), to learned (L152), Our (Table 1), double citation of the same reference (L234), the cross-domain tasks in Table 3 should be: Omniglot -> EMNIST & miniImageNet -> CUB.
- It is not clear from the text what network did you use to obtain the results in Table 1.


[1] Titsias, M. (2009, April). Variational learning of inducing variables in sparse Gaussian processes. In Artificial intelligence and statistics (pp. 567-574). PMLR.
[2] Achituve, I., Navon, A., Yemini, Y., Chechik, G., & Fetaya, E. (2021). GP-Tree: A Gaussian Process Classifier for Few-Shot Incremental Learning. arXiv preprint arXiv:2102.07868.
[3] Wilson, A. G., Hu, Z., Salakhutdinov, R., & Xing, E. P. (2016, December). Stochastic variational deep kernel learning. In Proceedings of the 30th International Conference on Neural Information Processing Systems (pp. 2594-2602).
[4] Bradshaw, J., Matthews, A. G. D. G., & Ghahramani, Z. (2017). Adversarial examples, uncertainty, and transfer testing robustness in Gaussian process hybrid deep networks. arXiv preprint arXiv:1707.02476.
[5] Wenzel, F., Roth, K., Veeling, B., Swiatkowski, J., Tran, L., Mandt, S., ... & Nowozin, S. (2020, November). How Good is the Bayes Posterior in Deep Neural Networks Really?. In International Conference on Machine Learning (pp. 10248-10259). PMLR.

**Time Spent Reviewing:**

8

---

> ### Author Response · Authors · 2021-08-10
> **Thank you for your constructive review!**
>
> Thank you for your careful review and constructive feedback. We address all your concerns in the following, and hope the responses would alleviate your concerns.
>
> **1. Classification likelihood**
>
> Thank you for bringing out the likelihood for discussion. In regression tasks with GP, $p(y|f)$ is a Gaussian likelihood. Whereas in multi-class classification tasks, we assume the discrete class labels $y$ follows a Categorical distribution. And the likelihood $p(y|\lbrace f^c \rbrace_{c=1}^C)$ is expressed by a softmax function, as in equation 11. The latent function $f^c$ is sampled from a posterior Gaussian distribution $q(f^c_*|x_*)$.
>
> ---
>
> **2. Sharing inducing inputs among new classes in testing**
>
> In meta learning, we usually assume an underlying task distribution, as discussed in [e]. While training and testing tasks does not share any classes, we see all tasks are sampled from this shared task distribution, thus by learning to learn a model capable of adapting to the tasks sampled from this distribution rapidly, testing tasks with unseen classes can be efficiently resolved with few samples.
>
> In our method, the inducing variables $\tilde{Z}$ are designed to be placed in the deep feature space, i.e., $dim(\tilde{Z})=dim(Z)$, rather than in the raw input space. The inducing variables are meta-learned to gain the knowledge across tasks within the underling task distribution, and adapt to new task through inner-loop adaptation. Intuitively, this shared knowledge exists and can be more easily learned in the high-level semantic space. Empirically superior performance reported in the paper confirms the effectiveness, and the adaptive dense inducing variables is even capable of handing cross-domain learning scenarios.
>
> ---
>
> **3. GPC references**
>
> Thanks for bringing those interesting references to us. For GP-Tree [a], it focuses on the few-shot class-incremental learning, which is not the scope of our paper. [b,c] are relevant to our work as they also work on Gaussian processes. We will include them in our paper for discussion.
>
> The recently proposed [a] introduces a model that expands to new classes with few samples through expanding a tree of binary GP heads. While both targeting a few-shot scenario, the contributions are orthogonal.
> While [b,c] discuss inducing variables for classification, they all focus primarily on supervised training, with inducing points trained end-to-end for the entire dataset. Our unique contribution is to allow adaptive inducing variables carrying prior knowledge about a distribution of tasks, and adapts to new tasks efficiently.
>
> ---
>
> **4. Implementation details and hyperparameter search**
>
> We would like to clarify that the temperature in the softmax function is meta-learned and updated in the outer-loop optimizations. Therefore our method does not require exhaustive search of temperature and the best temperature can be automatically optimized through data-driven training. Similar optimization driven hyperparameter selection is also shown to be effective in [d].
> We do not perform exhaustive hyperparameter search other than what have been presented in the ablation study.
>
> For miniImageNet, our model is trained for 100 epochs, with learning rate drop by 0.1 at the 40th, 60th epoch. For tieredImageNet,
> We will provide more implementation detail in The implementation will be release publicly upon acceptance.
>
> ---
>
> **5. Thank you for your thorough and careful reviews. All typos will be corrected in the revision.**
>
> For fair comparison, results in Table 1 are all obtained with the Conv-4 architecture, and we will clarify this in both the caption and the main text of the revision.
>
> ---
>
> [a] Achituve, I., Navon, A., Yemini, Y., Chechik, G., & Fetaya, E. (2021). GP-Tree: A Gaussian Process Classifier for Few-Shot Incremental Learning. arXiv preprint arXiv:2102.07868.
>
> [b] Wilson, A. G., Hu, Z., Salakhutdinov, R. R., & Xing, E. P. (2016). Stochastic variational deep kernel learning. Advances in Neural Information Processing Systems, 29, 2586-2594.
>
> [c] Bradshaw, J., Matthews, A. G. D. G., & Ghahramani, Z. (2017). Adversarial examples, uncertainty, and transfer testing robustness in gaussian process hybrid deep networks. arXiv preprint arXiv:1707.02476.
>
> [d] Bertinetto, L., Henriques, J. F., Torr, P. H., & Vedaldi, A. (2018). Meta-learning with differentiable closed-form solvers. arXiv preprint arXiv:1805.08136.
>
> [e] Finn, C., Abbeel, P., & Levine, S. (2017, July). Model-agnostic meta-learning for fast adaptation of deep networks. In International Conference on Machine Learning (pp. 1126-1135). PMLR.

---

> > ### Comment · Reviewer_jkmZ · 2021-08-16
> > **Follow Up Questions**
> >
> > I thank the authors for their response. Some details are more clear now, nevertheless, I still have a few questions:
> >
> > 1. Regarding the likelihood. I noticed the softmax likelihood mentioned in the paper. However, with the sotmax likelihood, the derivation in Appendix A (which was taken from the paper and technical report of Titsias (2009)) does not make sense as the derivation assumes a Gaussian likelihood. The authors used the loss obtained from this derivation to learn the inducing variables, but it is not clear why is that a correct thing to do and how exactly it was done.
> >
> > 2. Regarding the inducing variables. Just to verify that I understand correctly, during testing, the inducing variables are initialized at random in the feature space *for each new task*, and only they are updated using the inner loop loss, i.e., no outer loop loss occurs, is that correct?
> >
> > 3. Regarding the temperature scaling. Thanks for the clarification. However, tuning this parameter is orthogonal to the proposed method and I don't think that the current comparison to baseline methods is fair. When possible, this parameter should have been learned for baseline methods as well (or searched over when not possible), at least for the top competitors.
> >
> > 4. Regarding the uncertainty quantification (general massage). From OVE PG-GP paper, the ECEs are quite different from the ones reported here. Why is that? How does your method perform compared to this method in that aspect?

---

> > > ### Author Response · Authors · 2021-08-17
> > > **Response to follow-up questions**
> > >
> > > We sincerely appreciate your follow-up feedback. We provide a response in the following. Please let us know if you have any further concerns.
> > >
> > > **1. Further concerns about the likelihood**
> > >
> > > In our first round response, we thought the question was about the outer-loop meta-learning step, where we use the samples from posterior predictive of the query data as logits, which are inferred from adapted GPs and perform only maximum likelihood learning. As stated in our first response, we adopt Softmax likelihood, which does not affect the inner-loop optimization driven by ELBO. Note that we can also apply Gaussian likelihood on one-hot encoded labels here, which is detailed below and is adopted in the inner-loop adaptation, but Softmax directly allows a probabilistic interpretation of prediction outcomes, and we observed empirically that it performs better.
> > >
> > > The ELBO is tailored specifically for the inner-loop adaptations, and we provide a detailed explanation below. We start the discussion with the binary classification problem. It can be treated as a regression problem, where the labels $y \in \lbrace0, 1\rbrace$ are viewed as the regression targets and assumed to be Gaussian distributed. With the *one-vs-rest* strategy, we can extend the regression model to solve the $C$-way multi-class classification. To be specific, as stated in Line 134-136, $C$ regressors are learned with the support set data, where the $c$-th one is trained to predict whether a sample $x$ belongs to class $c$ (with target $y=1$) or not (with target $y=0$). We formulate each regressor as a single GP with its own inducing variables, and the Gaussian likelihood is adopted in the ELBO, as derived in Appendix A, to make each regressor learn to classify the corresponding class against the rest.
> > >
> > >
> > > **2. Test-time adaptation of inducing variables**
> > >
> > > As presented in L182 and Algorithm 1, in both training and testing, the inner-loop inducing variables are initialized with $\tilde{\mathbf{Z}}^0$ which are part of the learnable parameters of our model, and updated in the outer-loop. After training, the inducing variables $\tilde{\mathbf{Z}}^0$ are obtained and then used for the initialization of new testing tasks, so the inducing variables are not randomly initialized in testing loops. And you are correct that in the testing stage, there is not outer-loop, and there are only inner-loop adaptations of the inducing variables.
> > > The dense inducing variables (learned initialization) specify a shared Gaussian process prior over prediction functions of all tasks, and the per-task adaptation is further performed by updating the inducing variables in inner loops.
> > >
> > > **3. Temperature Scaling**
> > >
> > > The other two GP-based methods [21,31] neither adopt softmax or meta-learning with inner and outer loops that allow learning of $\tau$.
> > > For your interest, to enable fair comparisons, we performed further experiments by fixing $\tau = 1.0$, which reduces to standard softmax.
> > > As shown in the following table, we did not observe noticeable performance changes comparing to meta-learned $\tau$ in outer loops. All numbers are obtained with the Conv-4 architecture.
> > >
> > > |  Dataset    |MiniImageNet 1-shot|MiniImageNet 5-shot| CUB 1-shot |  CUB 5-shot |
> > > |-------------|:---------------:|:---------------:|:---------------:|:---------------:|
> > > | Ours w/ $\tau$| 56.32 $\pm$ 0.28  |  72.64 $\pm$ 0.26 | 70.14 $\pm$ 0.41 | 81.48 $\pm$ 0.58 |
> > > |Ours w/o $\tau$| 55.89 $\pm$ 0.28  |  72.32 $\pm$ 0.25 | 70.07 $\pm$ 0.42 | 81.05 $\pm$ 0.55 |
> > >
> > >
> > > **4. Uncertainty Qualification**
> > >
> > > We reported the numbers in percentage by following the convention of Table in [21]. Therefore the magnitudes of the reported numbers are different comparing to those in OVE PG-GP paper [31]. We have revised the table we presented in the *Few-shot regression and uncertainty quantification* above to include the numbers reported by [31].
> > > Generally, our method delivers improved uncertainty quantification comparing to most of the few-shot learning methods and the uncertainty quantification is comparable to other GP-based methods.

---

> > > > ### Comment · Reviewer_jkmZ · 2021-08-18
> > > > **Response to authors comment**
> > > >
> > > > I thank the authors for the detailed response. I think that the points discussed here should be clarified in the revised version of the paper as well. Based on the author's comments I decided to raise the score from 5 to 6.

---

> > > > > ### Author Response · Authors · 2021-08-19
> > > > > **Thank you!**
> > > > >
> > > > > All the additional discussions will for sure be included in the revision.
> > > > >
> > > > > Thank you for your support on our response.

---

> > > > ### Comment · Reviewer_JcrC · 2021-08-19
> > > > **Why does temperature affect accuracy?**
> > > >
> > > > Shouldn't the temperature parameter in the results above have no impact on the overall accuracy since temperature only affects the entropy/uncertainty and doesn't affect the actual class prediction? Do the authors have an explanation for this?

---

> > > > > ### Author Response · Authors · 2021-08-19
> > > > > **Temperature**
> > > > >
> > > > > Thanks for your further comment.
> > > > >
> > > > > The temperature here is not the same temperature adopted in calibrating the uncertainty quantification, which does not affect the accuracy.
> > > > >
> > > > > As shown in Eq. 11, we introduce temperature $\tau$ as a trainable parameter in the network, which is also adopted in [6]. As the value of $\tau$ is involved in the calculation of loss and back propagation, it will affect the results of the final model.
> > > > >
> > > > > We will further clarify this in the revision. Thank you.

---

### Official Review · Reviewer_JcrC · 2021-07-12

**Rating:** 6
**Confidence:** 4

**Summary:**

The authors propose a solution to few shot learning by using Gaussian processes with dense inducing points from the perspective of few shot learning, which allows for learning a shared set of inducing points which are then specialized into task specific inducing points.

**Limitations And Societal Impact:**

There is no mention of limitations or societal impact.

**Main Review:**

## Pros

- The results are strong and outperform all baselines methods.
- The method build off of work in GP's and meta learning
- The authors do a good job of citing past work in both GP and meta learning literature.

## Cons

- L40 (about GP's) ... "without suffering from computational cost in regular learning tasks" --> What does this mean? There is no computational cost? Even if you are saying there is reduced computational cost, is this even true? GP's scale as $\mathcal{O}(n^3)$
- Equation 7 has the order of arguments for the variational lower bound changed from equation 6. Is this a typo?
- The paper mentions dense inducing variables many times, but the inducing variables are actually quite sparse w.r.t. the task distribution. They are only dense when considered against the typical few shot learning task sizes. I think this distinction should be made clearer.
- L299 says that figure 3 indicates the efficiency of the proposed solution. This is a very bold claim that lacks evidence. One could easily also say that it indicates the method overfits easily to tasks, as the performance drops quickly with more inner steps. This overfitting is something that other methods such as [1] do not show even with large numbers of adaptation steps.
- There is no analysis of the added runtime for training/inference, as well as trainable variable counts. This information should be included somewhere in order to better understand the relation to other methods. The method leads me to suspect that there would be a minimal parameter increase, but a large increase in training/inference times.
- The authors don't describe any limitations of the work.
- The authors do not apply the problem to the very common few shot regression tasks. Although these are usually simple, I think the GP aspect make a particularly compelling use case for regression tasks which should be discussed.

My score is reflective of mostly bullet points 5 and 7 which need to be addressed in order to make the submission stronger.

 ## Minor

- L14 significantly --> significant
- L28 progresses --> progress
- L29 the few shot learning --> few shot learning
- L35 "widely developed" --> What does "widely" developed mean?
- L50 "we learning to learn" --> we learn to learn
- L73 "learning new task" --> learning new tasks
- L131 "few gradient descent update" --> few gradient descent updates
- L148 "an learned" --> a learned
- L154 "achieves better prior fitting the novel dataset" --> sentence needs to be rewritten to make sense

## References

[1] Nichol, A., Achiam, J., & Schulman, J. (2018). On first-order meta-learning algorithms. arXiv preprint arXiv:1803.02999.


**Time Spent Reviewing:**

4-5 hours

---

> ### Author Response · Authors · 2021-08-10
> **Thank you for your insightful feedback!**
>
> Thank you for your thorough review and insightful comments. We hope the responses below could address your concerns thoroughly.
>
> **1. L40 Computational cost**
>
> We regret for this imprecise statement. The discussion at L40 was made based on 'data scarcity' in few-shot learning settings. The small amount of labeled samples in each task does not introduce heavy burden to GPs at all as $n$ is usually small. This is contrast to regular image classification learning, where there is usually a large amount of data. We will revise our wording in the revision.
>
> ---
>
> **2. Equation 7**
>
> Yes, the different order of arguments was a typo, and we will correct this in the revision. Thank you.
>
> ---
>
> **3. Dense inducing variables**
>
> Thank you for mentioning this to us. Yes, indeed, dense inducing points are named in contrast to the typical size of the support set in few-shot learning tasks.
> We use inducing variables of a much larger size of the support set to carry the prior knowledge across tasks within a distribution.
> We will further clarify this in the revision.
>
> ---
>
> **4. Inner-loop steps**
>
> Thank you for this insightful comment. We will modify our statement on the efficiency of the proposed method based on the findings in Fig. 3. Our method can achieve good adaptation by a few inner-loop steps
>
> Please note that quick performance drop basically only happens when adopting unusually large inner-loop learning rate $\alpha=0.1$, and the performance is dropped from $79.2\%$ to $72.3\%$. Order settings have high tolerance to the number of steps. Moreover, the Y-axis might have exaggerated the performance drop, while such drops are actually smaller comparing some variants of [a] according to Figure 4 (a) of [1].
>
> ---
>
> **5. Few-shot regression experiments**
>
> Please see our response to all reviewers above.
>
> ---
>
> **6. Limitations**
>
> Please see our response to all reviewers above.
>
> ---
>
> [a] Nichol, A., Achiam, J., & Schulman, J. (2018). On first-order meta-learning algorithms. arXiv preprint arXiv:1803.02999.

---

> > ### Comment · Reviewer_JcrC · 2021-08-16
> > **Question about architectures**
> >
> > Thank you for your responses. I was looking through the paper again and I noticed that there is no label on which architecture was used in the experiments of Table 1 and 3. Which architecture was used for these experiments?

---

> > > ### Author Response · Authors · 2021-08-16
> > > **Clarifying architectures used in experiments**
> > >
> > > Thank you for your feedback. Following common practice and to enable fair comparisons, we adopt the standard 4 layer convolutional networks with 64 channels per layer (denoted as Conv-4) in all experiments if not otherwise specified. So all number in Table 1 and Table 3 are obtained with Conv-4. We will further clarify this in the revision.

---

### Official Review · Reviewer_PL2R · 2021-07-17

**Rating:** 6
**Confidence:** 4

**Summary:**

This paper presents a meta-learning algorithm based on Gaussian processes. The meta-learning or few-shot image classification algorithms based on Gaussian processes have been developed recently due to their principled uncertainty modeling mechanism, which can be particularly useful for few-shot learning settings where data are scarse. The core idea of the presented algorithm is its use of dense inducing variables; usually, the inducing point methods for GPs are employed when we have large training data so the direct computation of predictive or marginal distribution is costly. Hence, the inducing points methods are usually set to be sparse because they are introduced mainly for a computational reason. On the other hand, This paper interprets the inducing points as a shared knowledge that can be transferred between tasks and proposes to encode such meta-knowledge into them. To this end, the proposed method uses a dense set of inducing points so that rich meta-knowledge obtained during meta-training can be embedded into the inducing point and can be used to quickly adapt to new tasks. The resulting algorithm is a typical gradient-based meta-learning where the goal is to maximize marginal likelihood in the inner loop and the predictive likelihood in the outer loop, and the algorithm is trained via episodic training. The proposed algorithm is validated on common benchmarks on few-shot image classification and demonstrated decent performance.

**Limitations And Societal Impact:**

The authors did not address the limitations and potential negative societal impact as shown in the checklist.

**Main Review:**

I enjoyed reading the paper. The paper is well-written and clear, so easy to follow. The motivation is well-set, and I like the idea of transferring knowledge through inducing points. In my opinion, the main message of the paper is the use of dense inducing points whose size is much larger than typical support sets, and I think this idea can be applied to other related approaches.

Although not explicitly related, but I'd like to point out one related work that is worth citing. Set transformer (Lee et al., 2019) introduces an inducing point-based self-attention block to process set-structured data and is applied to meta-learning problems such as amortized clustering. There, the authors showed that their method with inducing points performs better than the full self-attention blocks without inducing points, and claim that the inducing points act as transferrable knowledge that can be shared among different tasks. I guess this view of interpreting inducing points is somewhat similar to the method proposed in this paper.

The experimental results are not that compelling; I'm not saying this because the tables in the paper do not list all the recent SOTA few-shot learning results (which would probably take an entire page). As the authors emphasized at the beginning of the text, meta-learning-based Gaussian processes provide a principled way of estimating uncertainties. In order to properly assess such aspects of GP-based meta-learning, I think the competing algorithms should be evaluated with the metrics related to uncertainty estimates (e.g., expected calibration error, negative log-likelihood, out-of-distribution detection performance, active learning, ...). It would have been good to have a minimal example on toy-regression tasks with meta-learned credible intervals displayed.

Another thing that needs to be checked is the complexity of the algorithm. Figure 2 reports that the performance starts to saturate with the number of inducing points $m=256$. Then the time-complexity of computing marginal likelihood is roughly $O(nm^2)$, which is quite heavy since $m \gg n$. How are the actual running times (training time and inference time) compared to the baselines? I guess the proposed method can be quite slower than the baselines without GPs (which just typically requires a few forward passes through feature extracting neural nets) and GP-based without inducing points ($O(n^3)$).

**Time Spent Reviewing:**

4 hours

---

> ### Author Response · Authors · 2021-08-10
> **Thank you for your thorough review!**
>
> Thank you for your positive and supportive comments. We hope the responses below could alleviate your concerns.
>
> **1. Set transformer**
>
> Thank you for bringing this very interesting paper to us, which we will include for our discussion.
> Yes, indeed, inducing points in Set Transformer [a] show the additional capacity of transferring shared knowledge across different tasks. This serves as as a supportive evidence for using inducing points in meta-learning. We use inducing points to collect and accumulate shared knowledge from previously seen, related tasks, which offer a strong inductive bias for efficiently and effectively solving new tasks with few labelled data. We will add this discussion in our revision.
>
> ---
>
> **2. Uncertainty quantification**
>
> Please see our response to all reviewers above.
>
> ---
>
> **3. Computation complexity**
>
> The size of our dense inducing points is relatively "large" to the size of a typical support set, which is up to 25.
> However, this does not introduce noticeable computation burden even comparing to methods with deep feature extractor only.
> Specifically, even when $m=256$, the induced computation roughly is not significantly noticeable and roughly equal to few additional linear layers.
> Given modern GPU, this additional time consuming is negligible.
> To show this, we present quantitative comparisons to some standard baselines in the following table.
>
> Methods|ProtoNet|MAML|DKT|Ours
> :---:|:---:|:---:|:---:|:---:
> Speed (s/task) | 0.015 | 0.092 | 0.018 |0.021
>
> All numbers are obtained with a ResNet-10 network on 5-way 5-shot CUB experiments.
> The speed of our method is comparable to those of the pure feature extractor based method (ProtoNet) and other GP based method (DKT), and is significantly faster comparing to MAML that requires inner-loop adaptations to all parameters.
> This table will be included in the revision.
>
> [a] Lee, J., Lee, Y., Kim, J., Kosiorek, A., Choi, S., & Teh, Y. W. (2019, May). Set transformer: A framework for attention-based permutation-invariant neural networks. In International Conference on Machine Learning (pp. 3744-3753). PMLR.

---

> > ### Comment · Reviewer_PL2R · 2021-08-25
> > **Thanks for the response**
> >
> > I appreciate the author's response which resolved most of my concerns. I keep my score intact.

---

### Official Review · Reviewer_yiLY · 2021-07-22

**Rating:** 6
**Confidence:** 2

**Summary:**

This paper develops a way to learn Gaussian processes with dense inducing variables by meta-learning for few-shot learning.
The algorithm provides a strong inductive bias for learning new tasks by specifying a shared Gaussian process prior over prediction functions of a set of tasks. It adapts the inducing variables to each task by efficient gradient descent to achieve task specific predictions.
Experiments show that this approach can achieve competitive performance in all the tasks considered, and state-of-the-art results in certain tasks.

**Limitations And Societal Impact:**

There is no discussion on the limitations and potential negative societal impact of the work.

**Main Review:**

1. The paper is relatively well written.
2. In terms of originality, I am not aware of other similar works.
3. The empirical results are pretty strong, achieving state-of-the-art results in certain settings.

**Detailed comments**:
1. Are there any theoretical guarantees for algorithm 1?
2. In equation 5, should it be $p(y|f_m)$ instead of $p(y|f)$?
3. Minor: some formatting issues at Line 20, 24.

**Time Spent Reviewing:**

5

---

> ### Author Response · Authors · 2021-08-10
> **Thank you for your supportive review!**
>
> We are grateful for your thorough review and valuable comments. We are glad that you appreciate the idea of this paper, and hope our responses will address your concerns fully.
>
> **1. Theoretical guarantee of Algorithm 1**
>
> We assume that the reviewer means theoretical guarantee for convergence. Our algorithm is in the family of gradient based optimization under the meta-learning setting. The gradient based inner-loop adaptation is guaranteed by several theoretical meta-learning works [1,2]. Our method employs maximization of evidence lower bound, which is further guaranteed by theory of variational inference.
>
> ---
>
> **2. Equation 5**
>
> In equation 5, we factorize the joint distribution $p(y, f, f_m)$ as, $p(y, f, f_m)=p(y|f, f_m)p(f, f_m)=p(y|f)p(f|f_m)p(f_m)$, which is based on the Markov Chain structure $f_m \rightarrow f \rightarrow y$, as illustrated in the graphical model in Figure 1. This structure indicates that the label $y$ does not directly depends on inducing variables $f_m$, as well as $p(y|f, f_m)=p(y|f)$.
>
> ---
>
> **3. Formatting issues**
>
> Thanks for pointing out the formatting issues. All these typos will be corrected in the revision.
>
> ---
>
> **4. Limitations and societal impact**
>
> Please see our response to all reviewers above.
>
>
> [1] Fallah, A., Mokhtari, A., & Ozdaglar, A. (2020, June). On the convergence theory of gradient-based model-agnostic meta-learning algorithms. In International Conference on Artificial Intelligence and Statistics (pp. 1082-1092). PMLR.
>
> [2] Balcan, M. F., Khodak, M., & Talwalkar, A. (2019, May). Provable guarantees for gradient-based meta-learning. In International Conference on Machine Learning (pp. 424-433). PMLR.

---

> > ### Comment · Reviewer_yiLY · 2021-08-26
> > **Thank you for the response**
> >
> > Thank you for the response! I would keep my score.

---

### Author Response · Authors · 2021-08-10
**Thank you all for your constructive comments!**

We thank all reviewers for the supportive and constructive comments. We will update our manuscript to include all the important discussions and results points by reviewers.

We first address the shared concerns here, and address the comments raised by each reviewer individually.

**1. Limitations and societal impact**

The number of inducing variables: Currently, in our method the number of inducing variables needs to be pre-fixed. This can be a potential limitation since we can only empirically set the number of inducing variables with no prior knowledge to rely on. As a rule of thumb, for more complex tasks, we might need a larger set of inducing variables to be able to collect and carry enough shared knowledge. We will add this discussion in the paper. In addition, we do not observe any foreseeable negative societal impact of this work.

Negative Societal Impact: As our method is a meta-learning method aiming at quickly adapting to new tasks, as well as we test our methods on standard object classification datasets, we do not think there are any particular negative societal impacts of our method.

---

**2. Few-shot regression and uncertainty quantification**

Due to the page limit, the natural disadvantages of applying GP on classification, and the relatively lower difficulties of common few-shot regression experiments, we removed the few-shot regression experiments and leave the space for the challenging in-domain and cross-domain few-shot classification experiments.
We show the results on regression tasks in figure below.

Anonymous link to figure: [Regression](https://github.com/Submission112358/fornips/blob/main/Presentation2.jpg)

Without tuning to the hyperparameters, our method achieves both good fits and uncertainty quantifications on the 10-shot sine wave regression.

As for the uncertainty quantification, we show in the table below the average Expected Calibration Error (ECE) with standard deviation over 3 runs on 1-shot and 5-shot classification (5-ways) in the CUB dataset. Each method is evaluated on 3000 randomly generated test tasks. Our method shows comparable results with others.

| Method      |      1-shot     |      5-shot     |
|-------------|:---------------:|:---------------:|
| MAML        | 1.14 $\pm$ 0.22 | 2.47 $\pm$ 0.07 |
| MatchingNet | 3.11 $\pm$ 0.39 | 2.23 $\pm$ 0.25 |
| RelationNet | 4.13 $\pm$ 1.72 | 2.80 $\pm$ 0.63 |
| DKT         | 2.62 $\pm$ 0.19 | 1.15 $\pm$ 0.21 |
|OVE GP ML    | - | 2.6 |
|OVE GP PL    | - | 0.5 |
| **Ours**    | 2.31 $\pm$ 0.23 | 1.43 $\pm$ 0.31 |

We will add the discussion on few-shot regression and uncertainty quantification in the revision.

---

### Decision · Program_Chairs · 2021-09-27

**Decision:**

Accept (Poster)

**Comment:**

The reviewers all agree that the idea of using a dense set of inducing points to share information across tasks in the few-shot setting is novel and interesting. The discussions yielded a number of clarifications as well as new experiments on few-shot regression and uncertainty quantification. Please be sure to add these to the final draft.